# Local colonisations and extinctions of European birds are poorly explained by changes in climate suitability

Climate change has been associated with both latitudinal and elevational shifts in species' ranges. The extent, however, to which climate change has driven recent range shifts alongside other putative drivers remains uncertain. Here, we use the changing distributions of 378 European breeding bird species over 30 years to explore the putative drivers of recent range dynamics, considering the effects of climate, land cover, other environmental variables, and species' traits on the probability of local colonisation and extinction. On average, species shifted their ranges by 2.4 km/year. These shifts, however, were significantly different from expectations due to changing climate and land cover. We found that local colonisation and extinction events were influenced primarily by initial climate conditions and by species' range traits. By contrast, changes in climate suitability over the period were less important. This highlights the limitations of using only climate and land cover when projecting future changes in species' ranges and emphasises the need for integrative, multi-predictor approaches for more robust forecasting.

Climate change is a major threat to global biodiversity, driving changes in the distributions[1] and abundances[2] of a wide array of taxa[3]. Yet, the extent to which climate change drives species' range shifts across large geographic regions in relation to other putative drivers of change, remains contested. Climate change has been associated with species' ranges shifting towards higher latitudes and altitudes[4–6], although more complex responses can occur[7]. These may be a consequence of non-directional and non-linear changes in climate (e.g., precipitation patterns). Range shifts may also result from interactions between climate change, changes in land use and persecution by humans, and other site-specific events[1,8–10]. The capability of species to track climate change may also be modulated by their adaptive capacity, with some traits potentially affecting species' ability to respond and moderate the potential impacts of climate change on their ranges[11–13]. For example, the behavioural and demographic characteristics of species, such as dispersal ability, fecundity, and habitat and diet specialisation, can facilitate or moderate changes in range[5,14–16]. Moreover, the interaction between various species' traits and extrinsic drivers of change may also lead to highly complex and idiosyncratic patterns of range shifts[17].

Two processes determine species' range shifts: colonisation and local extinction[8,10,17]. Climate-driven colonisation is likely to occur where climatic change has directly or indirectly released species from barriers to dispersal and population establishment, whilst local extinctions are likely to occur where climatic change has made conditions unsuitable for the local persistence of a species[18,19]. Colonisations may lag behind changes in climatic suitability because other habitat requirements, for example prey availability, are not yet met in newly climatically suitable areas, or there may be insufficient emigrating dispersive individuals to colonise new areas effectively. Extinction debts may accrue where the climate becomes unsuitable, but the proximate determinants of occupancy persist for some time[10]. Understanding the mechanisms that underlie species' range shifts and establishing the relative roles of climate change and other putative drivers of range change is therefore critical if we are to identify species' extinction risk and implement targeted conservation actions. The roles of extrinsic environmental conditions and species' traits in driving colonisation and local extinctions at a continental scale, however, remain largely unexplored.

✉e-mail: christine.howard@durham.ac.uk; s.g.willis@durham.ac.uk

Here, we use the data from two European breeding bird atlases separated by a 30-year period to test the relative roles of different putative drivers of local extinction and colonisation events[20,21]. The data for the first atlas (1985–1988) were collected at the onset of a period of major climatic change across Europe (Supplementary Fig. S1). The publication of the second atlas (data collected 2013–2017) provides a unique opportunity to quantify changes in species ranges at a continental scale over three decades of substantial environmental change. These data also provide an excellent opportunity to test the extent to which the projections from species distribution modelling (SDMs) have been realised[22,23]. SDMs are a widely used tool for predicting the potential impacts of climate change on species' distributions[24]. However, they have limitations, often failing to account for other environmental constraints, biotic interactions, species' adaptive capacity or dispersive abilities[25,26]. By comparing observed range shifts with those predicted from SDMs, we can quantitatively assess the importance of these apparent limitations. The predictions of SDMs have been tested on national[27,28] and continental scale datasets[29–31], demonstrating that SDMs can often fail to predict observed species' range shifts under climate change. However, the extent to which climate change has driven observed range shifts in relation to a suite of other biotic and abiotic factors, such as land cover change, proximity to source populations, and species traits still needs to be assessed.

We use data on long-term changes in the breeding ranges of 378 species of European birds to quantify the influence of a suite of environmental covariates, including climate and land cover, and species' traits, on changes in species' ranges. Specifically, we address three key questions: (1) has climate change been the major driver of range shifts across Europe in recent decades? (2) how do other environmental factors, such as land cover, and species' traits influence observed species' range changes? And (3) do the drivers of colonisation and local extinction events differ? To do this, we first use SDMs to predict the change in the breeding ranges of 378 species of European breeding birds between 1985–1988 and 2013–2017 expected from either climate change only, or from climate and land cover change, which we compare with the observed range shifts over the same period. We then assess the importance of a suite of environmental covariates and species' traits in causing local colonisation and extinction events. By analysing colonisations and extinctions separately, we aim to advance understanding of the processes underlying observed range shifts. If change in climate suitability is the predominant driver of observed range shifts, we expect colonisation and local extinction events to occur at the leading and trailing margins of a species' climatic niche respectively. If there is no effect of climate change on species' range shifts, we expect that local dispersal processes would lead to colonisations centred in and around areas of high probability of occurrence (and which we use to define areas of high initial climate suitability) while local extinctions would occur in areas with lower initial probability of occurrence (and therefore estimated as being of lower initial climatic suitability). If species' range shifts are driven by other environmental processes or species' traits, or the interaction between these factors, we expect more diverse patterns of range shifts (See Supplementary Fig. S2 for more details on our hypotheses).

Our results show a lack of congruence between species' observed range shifts and those expected from either climate only or climate and land cover SDMs, in terms of both direction and rate. Specifically, we find that the directions of observed range shifts are more variable than predicted by SDMs. We find that proximity to source populations and the underlying climatic suitability of an area are of the greatest importance in determining the occurrence of colonisation and local extinction events. By contrast, we find the role of changing climate suitability over the 30-year period, although significant, to be of lesser importance. We demonstrate that, despite major differences in the processes of species' colonisation and extinction, the underlying

predictors are broadly similar. These results suggest that recent observed changes in the ranges of European breeding birds have not been predominantly driven by changes in climate, but rather have been moderated by other extrinsic environmental factors and species' traits. This has important implications for our understanding of, and our ability to predict the impacts of, future climate change on biodiversity.

## Results and discussion

### The importance of climate change for driving species' range shifts

Between 1985–1988 and 2013–2017, the European breeding ranges (measured as the number of occupied 50 km × 50 km grid cells) of 140 of the 378 study species expanded by >5%, whilst the ranges of 110 study species contracted by >5%. The median observed shift in the centre of gravity (COG) of species' ranges was 71 km (95% CIs = 6–638 km) in a median direction of 6°N but with high variance (95% CIs = −173°S–173°S). The median rate of shift in species' COG was 2.4 km/yr (95% CIs = 0.2–21.2 km/year, Fig. 1a and Supplementary Fig. S3). When comparing the COG of observed range shifts to those projected by climate-only SDMs, we saw significant differences in both the direction (Watson–Wheeler test: $W = 154.59$, df = 2, $p < 0.01$) and magnitude (Wilcoxon Signed Rank test: $V = 30057$, $p < 0.01$) in the shift of the observed and projected COG (Fig. 1, Supplementary Figs. S3 and S4). Projected range shifts were in a median direction of −0.9°N, with much lower variance than observed range shifts (95% CIs = −141.6°SW–127.1°SE). Predicted range shifts were at a median rate of 3.4 km/yr (95% CIs = 0.5–11.3 km/yr, Fig. 1b). We also found significant differences in both the direction (Watson–Wheeler test: $W = 129.16$, $p < 0.01$) and magnitude (Wilcoxon Signed Rank test: $V = 45489$, $p < 0.01$) in the projected shift of the COG (Fig. 1c, Supplementary Fig. S3) when comparing the COG of observed range shifts to those projected using SDMs fitted with both climate and land-cover variables. Projected range shifts from SDMs fitted with climate and land cover variables were in a direction of −7.9°N (95% CIs = −151.1°SW–96.9 °N) at a median rate of 2.2 km/yr (95% CIs = 0.3–7.3 km/yr). In short, neither the 'climate-only' or the 'climate and land cover' informed SDMs predicted the median COG range shifts well, with the observed data also showing much higher variability in the direction and pace of observed shifts.

### The abiotic and biotic predictors of range shifts

To develop a fuller understanding of the processes that lead to species' range shifts, we assessed colonisation and local extinction events separately. We used generalised linear mixed effects models to quantify the relationships between the occurrence of colonisation or extinction events and a suite of environmental variables and species' traits. These models revealed that, on average across species, slightly more colonisations occurred where climatic suitability increased, and more extinctions occurred where climatic suitability declined (Fig. 2). Some species, such as Brambling (*Fringilla montifringilla*, Fig. 3c, d), tracked projected changes in climate suitability between the two periods very well, but many other species did not. Our models revealed that the initial climatic suitability for 1985–1988 was, overall, a more important predictor of colonisations and extinctions over the following 30 years than change in climate over that period (Fig. 2). In particular, we found that extinctions were more likely in areas where our estimates of initial climate suitability were low, and colonisations were more likely in areas where estimated initial climate suitability was high. As it is almost axiomatic that climate determines species native ranges[24], this may reflect a true signal of the importance of initial climatic suitability. For example, environmental changes in an area may not be sufficient to ensure the persistence of a species in areas of already very low climatic suitability. Alternatively, modelled estimates of the importance of climatic variables may, in part, reflect underlying

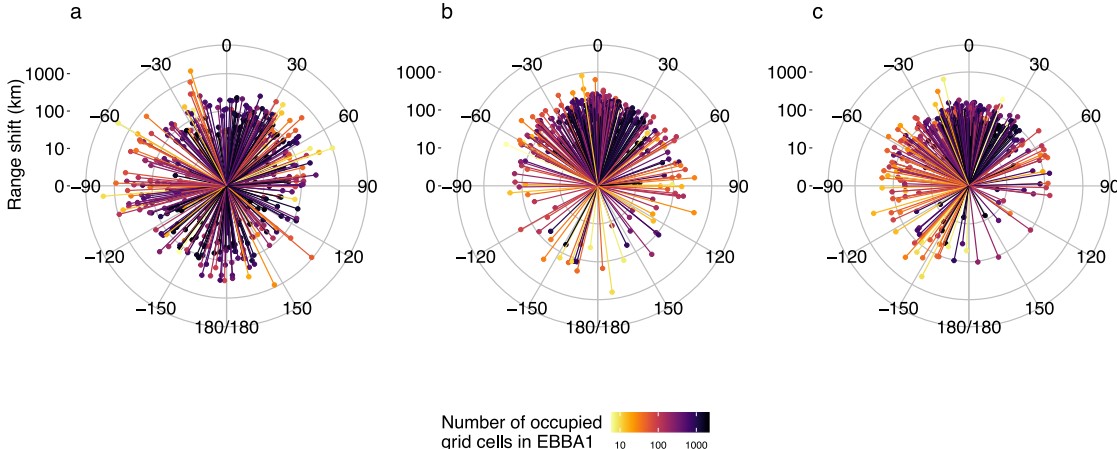

**Fig. 1 | Observed (a) and predicted (b and c) shifts in the distance and direction of the ranges of 378 species of European breeding birds between the periods 1985–1988 and 2013–2017.** Observed range shifts (**a**) are based on the species' range data from the European Breeding Bird Atlases (EBBAs), whilst predicted range shifts are from species distribution models fitted to species data and either climate only (**b**) or climate and land cover (**c**) data related to the 1985–1988 atlas and projected to the 2013–2017 period. Each line represents a single species. The centre of each polar plot represents either the observed (**a**) or predicted (**b**, **c**) centre of gravity of the 1985–1988 range for each species. Lines show the distance and direction between the observed or predicted 1985–1988 centre of gravity and the equivalent 2013–2017 range. Lines are coloured by the number of 50 × 50 km UTM grid cells a species occupied in 1985–1988 (i.e., EBBA1). Source data are provided as a Source Data file.

spatially autocorrelated factors not considered in typical SDM models, including biotic interactions, adaptive evolution and historic events[25]. This would not necessarily imply that climate does not determine species' distributions, but rather, estimates of the species-climate relationship are less robust. Importantly, this does not undermine our findings of the limited influence of change in climate suitability. As our measure of climate change is based on observed, rather than predicted data, we would also expect changes in climate to correlate with changes in species' ranges where there is a causal relationship.

The occurrence of colonisation and extinction events were also strongly influenced by the distance from the nearest continually occupied 50 × 50 km grid cell, with colonisations more likely to occur closer to occupied grid cells, and extinctions more likely in more distant areas. This pattern matches the expectations of the 'rescue effect'[32], with less isolated patches likely to receive more dispersing individuals resulting in either a greater probability of colonisation or the 'rescuing' of more susceptible populations from becoming locally extinct (Supplementary Fig. S2). We also found that, in areas closer to the centre of a species range, colonisation events were more likely, whilst extinction events were less likely. In areas closer to a species range centre, often a larger fraction of suitable habitat is occupied, producing a greater abundance of emigrating dispersive individuals, and further increasing the probability of colonisation or the 'rescuing' of smaller populations[32]. Models of colonisation and local extinction events fitted without these two variables (distance to the nearest continually occupied grid cell and the distance of a grid cell to the species COG in the first atlas) performed substantially worse than those fitted with them (Models fitted with distance variables: colonisation models mean marginal $R^2 = 0.59$, S.D. ± 0.03, extinction models mean marginal $R^2 = 0.82$, S.D. ± 0.02; Models fitted without distance variables: colonisation models mean marginal $R^2 = 0.17$, S.D. ± 0.02), extinction models mean marginal ($R^2 = 0.55$, S.D. ± 0.03). These results provide further evidence of the strong influence that the interactions between spatially structured populations can have on species' range dynamics[32]. Furthermore, they also suggest that the ability to predict species future range shifts without considering which cells are continuously occupied would likely be poor.

Larger areas of suitable habitat also increased the likelihood of colonisation and reduced the likelihood of extinction. Notably, species associated with forest habitats, such as the Middle Spotted Woodpecker (*Leiopicus medius*, Fig. 3e, f), appear to have benefitted from the widespread expansion, conservation and maturation of forests across much of Europe[33]. Conversely, despite large-scale improvement in simulated climate suitability across Europe, some species associated with agricultural environments, such as the Crested Lark (*Galerida cristata*, Fig. 3a, b), have undergone substantial population declines across some parts of their range in response to widespread patterns of agricultural intensification and abandonment[34]. It has been suggested that there is a hierarchy of environmental controls on species' distributions, with climate operating at the largest scales, and topography and land cover moderating these effects at smaller scales[35]. Despite this, our results show that land cover still has a strong and significant effect in driving colonisation and local extinction events, even at a 50 × 50 km scale.

Species' traits had some influence on the likelihood of colonisation and extinction events. Species that have either a history of persecution and/or are listed on Annex I of the European Union (EU) Birds Directive, for which the Member States are obliged to implement special conservation measures, were less likely to undergo local extinctions and more likely to colonise new areas. For example, the White-tailed Sea Eagle (*Haliaeetus albicilla*, Fig. 3g, h) has undergone substantial range expansion over the past 30 years in response to a reduction in persecution and numerous reintroduction programs[36]. Many of the species listed on Annex I and previously subject to persecution, such as birds of prey, may also have benefited from reductions in pesticide use since the 1970s[37]. For example, the thickness of European Sparrowhawk (*Accipiter nisus*) eggs in the UK (and consequently productivity) did not recover until DDT was banned in 1986[38]. This lower rate of local extinctions of Annex I species provides further support for the positive impact of EU conservation legislation on target species[39].

Species with larger ranges had higher probabilities of local extinctions (Fig. 2), despite the expectation that narrowly distributed species are more vulnerable to extinction[32,40]. Notwithstanding the influence of range size, most species-specific traits were of limited importance in our models. This aligns with previous studies that also found species' traits to be poor predictors of range shifts[41]. This is likely due to many hypothesised confounding effects of species' traits on range shifts. For example, species with greater diet breadth are expected to shift their ranges more readily as they are more likely to

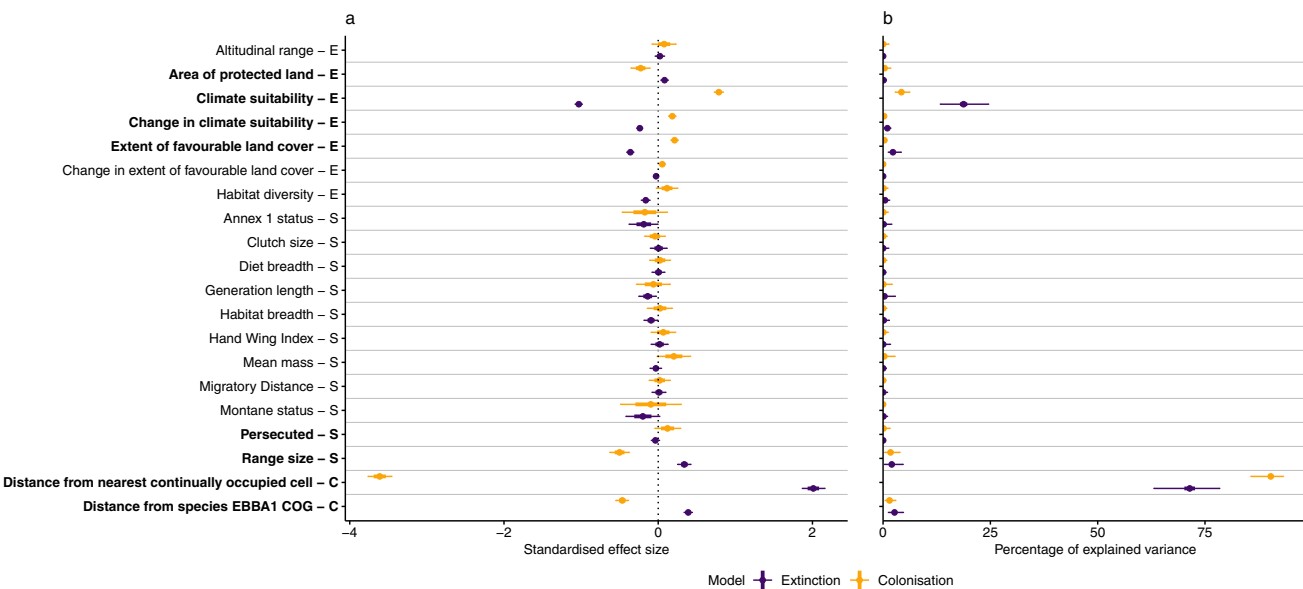

**Fig. 2 | Standardised coefficients (a) and percentage of variance explained (b) from MCMC generalised linear mixed models of the colonisation (orange, mean marginal $R^2$ = 0.59, S.D. ± 0.03) and extinction (purple, mean marginal $R^2$ = 0.82, S.D. ± 0.02) events of 336 species of European breeding birds between 1985–1988 and 2013–2017.** Fewer species were included in this analysis due to the completeness of explanatory variables. Variables are grouped into broader classes, which are indicated by the capital letters on the side of the variable names: Environmental covariates (E), Species-specific traits (S), and Species cell traits (C). In **a**, points indicate the mean estimated effect size, thick horizontal bars indicate the posterior standard deviations, and the thin horizontal lines indicate the 95% credible intervals of the coefficient values produced by averaging 100 separate MCMCglmms. Emboldened y-axis labels indicate variables for which their CIs did not overlap zero in either set of models. To standardise coefficient values, all predictors were z-transformed. In panel b, points indicate the median, thick horizontal bars indicate the inter-quartile range, and the thin horizontal lines indicate the 95% confidence intervals of the percentage of explained variance across 100 separate MCMCglmms. Source data are provided as a Source Data file.

find suitable food in novel environments, yet specialist species are also likely to demonstrate more rapid range shifts as they track their resources[14,42,43]. Single measures of species' traits also do not reflect interspecific variation across geographic ranges and/or phenotypic variation and plasticity over an organism's lifetime. It may be that traits associated with specific populations or life-history stages may be more strongly related to the processed driving range shifts, e.g. the propensity for birds living at the leading edge of their range to be larger[44]. Treating species as fixed entities, may not encapsulate the trait values responsible for driving range shifts[41,43].

Despite the fundamental differences in the processes of colonisation and extinction events[45], our results revealed marked similarity in the putative drivers of these events; distance to the nearest continually occupied grid cell, climate suitability and range size were the three most important variables in our models of colonisations, whilst the three most important variables in our models of extinctions were distance to the nearest continually occupied grid cell, climate suitability, and distance to species' COG (Fig. 2b). Surprisingly, given that species are theorised to be more susceptible to changes in climate at their poleward range margin than at their equatorial range margin (where inter-specific competition has been proposed as an important range-margin determinant)[18], we found no evidence for differences in the importance of change in climate suitability between the two sets of models. Furthermore, changes along species' trailing range margin are thought to be modulated by their population dynamics, with longer-lived, less dispersive species expected to respond more slowly[43,46]. However, we found no evidence that species' traits were stronger drivers of local extinctions than colonisations.

### Wider implications
The lack of congruence between observed range shifts and those predicted by climate SDMs, and the small effect of change in climate suitability in our models of colonisation and extinction events, suggest

that recent changes in climate were not the predominant driver of shifts in bird species' ranges across Europe since the 1980s. However, we stress that this finding does not necessarily suggest that climate is an unimportant driver of species' ranges, nor that species are not shifting their ranges in response to climate change. Our analysis does reveal a small, significant effect of change in climate suitability on species' colonisations and extinctions. Furthermore, any climate-driven changes in local population density that do not result in either a colonisation or an extinction event will not be reflected in our results. Changes in climate have been widely associated with species' population trends[2,47]. In our analyses, a species that had undergone large declines in local abundance would still be regarded as present if it persisted, even if much reduced. Moreover, if the proximate determinants of occupancy allow a species to continue to persist in an area despite local declines in climate suitability, climate-driven extinction debts may accrue[10]. Despite the relatively long study period, species' range responses may lag behind the rate of climate change, limiting our ability to detect the true importance of changes in climate in driving species' range shifts. Finally, despite bioclimatic variables often being strongly correlated, it may be that species are responding to aspects of climate not reflected in our SDMs[48].

Our results emphasise that species' range responses to climate change are likely complex, modulated by both other extrinsic environmental factors and species' traits. The substantial variation in observed species' range shifts is striking. Studies have previously shown that species' range shifts in response to climatic change are often idiosyncratic[1,4,49], and rarely closely mirror expectations of predicted range shifts under climate change[27,29,30]. Here, we have also shown that the ability of SDMs to predict species responses to climate change is not improved by also accounting for changes in land cover. Our results demonstrate how recent climate change often contributes relatively little to contemporary range changes. Instead, initial climate suitability, other environmental constraints such as the extent of

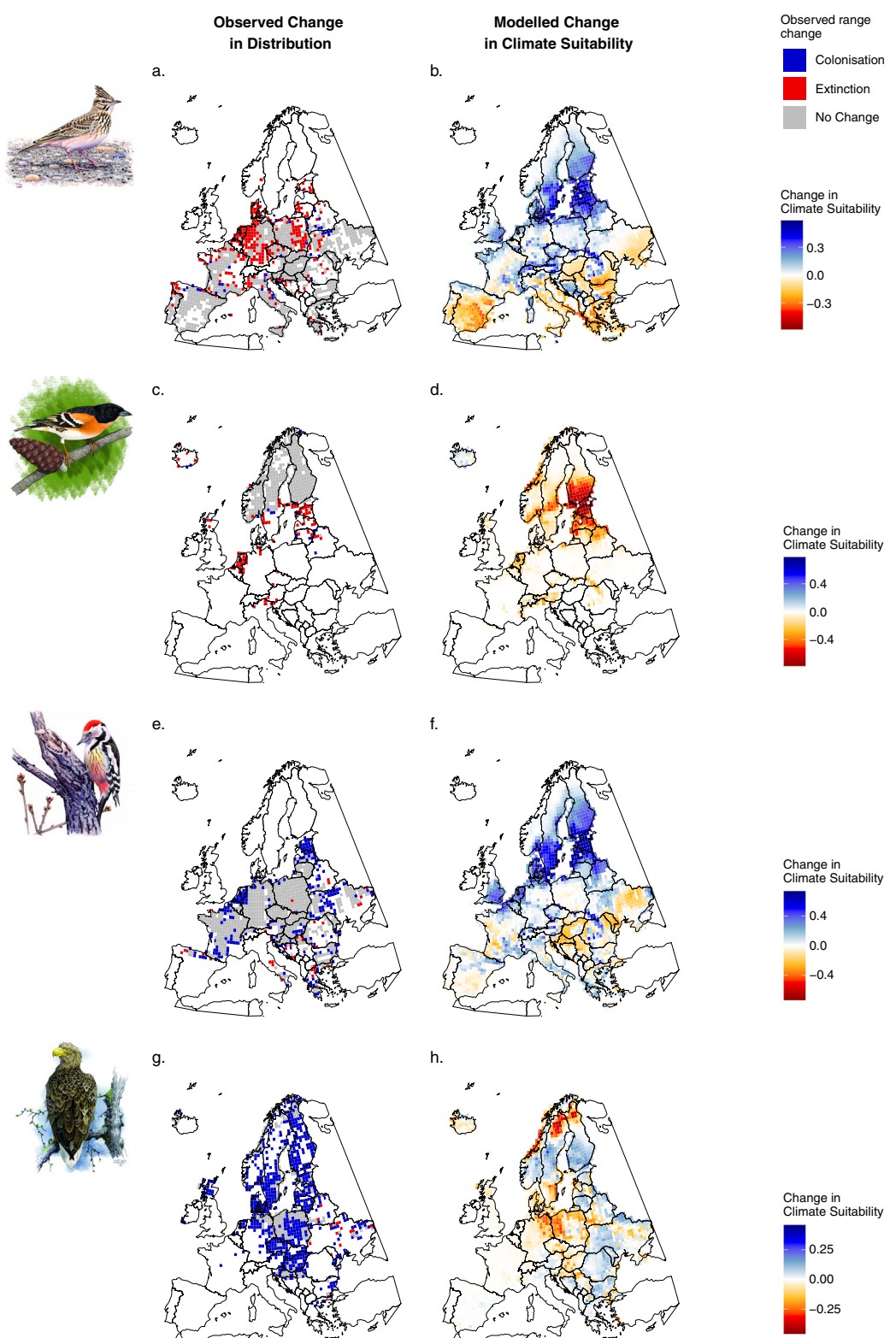

**Fig. 3 | Observed changes in species' ranges between 1985–1988 and 2013–2017 (a, c, e, and g) and predicted changes in climate suitability for the same time period (b, d, f, and h) for four species of European breeding birds. a**, **b** *Galerida cristata* (image by Szabolcs Kókay)*,* **c***,* **d** *Fringilla montifringilla* (image by Lluís Sogorb)*,* **e**, **f** *Leiopicus medius* (image by Szabolcs Kókay), and **g**, **h** *Haliaeetus albicilla* (image by Eugeny Koblik). Maps of observed changes in species' ranges (left-hand maps) indicate areas where species have colonised (blue), areas where species have gone locally extinct (red) and those areas which species occurred during both Atlases (grey) between 1985–1988 and 2013–2017. Maps of predicted changes in climate suitability (right-hand maps) show the difference in the predicted climate suitability between the two time periods. Climate suitability was calculated using species distribution models based only on bioclimatic variables. Note, the colour scale of climate suitability plots varies between species. Source data are provided as a Source Data file.

favourable land cover, and species range traits explain some of this variation. Crucially, our finding of the importance of the proximity to source populations highlights the role that the interactions between spatially structured populations can have on species range dynamics, and the value of accounting for metapopulation processes when anticipating range shifts. Thus, we conclude that predictions of species' range shifts, as a response to future environmental change, based on climatic and broad-scale land cover variables are unlikely to be able to fully capture these complex responses. This has important ramifications for interpreting studies that have predicted species' range shifts based only on species' climatic niches[22,23]. Our results emphasise the importance of considering the other complex processes that can drive colonisation and extinction events, alongside extrinsic environmental factors, for robust forecasting of species' range responses to future environmental change.

## Methods

### Species data

Data on the distributions of European breeding birds were obtained from two Europe-wide distribution atlases[20,21]. These atlases provide records of each species' occurrence across Europe in circa 2819 50 × 50 km squares of a modified Universal Transverse Mercator (UTM) grid. Data in the first atlas were collected during the 1980s (mainly 1985–1988), and in the second during the 2010s (mainly 2013–2017), i.e., around three decades after the first. For a species to be considered present in a grid cell, at least one pair of breeding individuals had to be recorded during the sampling period. Further details on the collection of these data can be found in Hagemeijer & Blair[20], and Keller et al.[21]. In Russia and some other parts of eastern Europe species records were primarily qualitative in one or both atlases and were therefore excluded from our analyses. Details on how regions with poor sampling coverage were identified can be found in Keller et al.[21]. Data from 2117 50 × 50 km grid cells were used in the analyses, which equates to a total study area of 5,292,500 km² (Supplementary Fig. S5). From the 625 species included in either of the two European atlases, some species were not included in the analyses for the following reasons. We excluded species considered as either introduced or invasive in Europe. We also excluded species that were considered to breed irregularly either in the first and/or the second atlas, and species that spend a significant proportion of their time at sea, whose ranges and abundances will be predominantly driven by processes not reflected in our variables. Taxonomic changes (i.e. species lumped or split) and changes in grid allocation between the two atlases were taken into account so that the dataset used in the analyses was consistent for a 30-year comparison of species occurrence across squares[21]. For example, *Phylloscopus bonelli* and *Phylloscopus orientalis* were considered to be the same species in the first atlas, so they were treated as a single taxon in our analyses. Due to model building limitations in SDMs, 36 species that were recorded in fewer than five grid cells were also excluded from further analyses. This left 378 species of European breeding birds for subsequent analysis (See Supplementary Data 1 for a list of species).

Species' trait data, for a suite of traits previously hypothesised to be informative for predicting species' range shifts[50], were collated for the 378 species from published data sources. These traits included mean body mass[51], generation length[52], clutch size[53], hand-wing index[54], migration status[55], and diet breadth[56]. Mean migratory distance was calculated as the great-circle distance between the centre of gravity of species' breeding and non-breeding ranges, as defined by BirdLife International[57] and following Gilroy et al.[58]. As there was a strong association between migration status and migratory distance (Supplementary Fig. S6), we repeated the analysis with each variable. There were no substantial differences between models fitted with either variable (Supplementary Figs. S7 and S8). Diet breadth was quantified as Shannon's richness index (calculated with R package

'vegan'[59]) of the proportional use of different diet categories: invertebrate, vertebrate (endotherm), vertebrate (ectotherm), fish, vertebrate (unknown), scavenge, fruit, nectar or pollen, seed, other plant material. We also collated information on the breadth of habitats used by a species from Ducatez et al.[60]. These data were, however, only available for 336 species. We, therefore, repeated the analyses described below, with and without this variable. The inclusion of this variable had no substantial effect on the overall results of the analysis (Supplementary Figs. S7 and S8). We also separately classified whether species were montane, as montane species have been hypothesised as being especially vulnerable to climate change[61]. We collated information on whether species were specially protected (Annex I status, EU 2009) or had been subject to persecution across Europe. For the latter, we obtained information for each species on the number of birds killed or taken illegally in Europe per year[62,63], which we then expressed as a percentage of the total size of the European population using information from BirdLife International[64]. Finally, we calculated the size of each species' range in the first atlas, measured as the number of occupied 50 × 50 km grid cells. See Supplementary Data 1 for all species-specific values.

### Environmental covariates

**Climate suitability.** To provide a measure of the change in climate suitability for each species in each 50 × 50 km grid cell, we used species distribution models (SDMs) to model each species' climatic niche across Europe, Turkey, and North Africa. The latter regions were included to encompass the southern range margins of as many species as possible, thus improving the overall performance of the SDMs[65]. For Europe, we used the occurrence data from the first atlas[20]. Occurrence data for Turkey and North Africa were obtained from BirdLife International and NatureServe[57]. These data are available as range polygons, which we intersected with the same 50 km × 50 km UTM grid used in the atlases.

To best capture the climatic niche for each individual species, we employed a model selection procedure to select a set of ecological relevant variables that are non-collinear and produce high-performing models[66]. Data for two climatic variables, mean monthly temperature and precipitation from 1968 to 2017 (the period during which most data underlying the species' range extent maps of both atlases were collected), were obtained from the CRU TS 3.25 0.5° dataset[67]. These data were used to calculate eight bioclimatic variables at the same resolution and for the same area as the species' range data. These included mean annual temperature and precipitation, seasonality of both temperature and precipitation, maximum temperature of the warmest month, minimum temperature of the coldest month, precipitation of the wettest month, and precipitation of the driest month. These variables capture the typical conditions on the breeding grounds of European breeding birds, along with the variability and extremes in those conditions, and have previously been shown to be informative in describing both the range extents and abundance patterns[47,68,69] of these species. We calculated mean values of these eight bioclimatic variables for two time periods, 1968–1988 and 1997–2017. The use of a 20–30-year period to represent the average 'climatic normal' conditions for an area is standard practice in the field of climate science, with these time periods overlapping when most species' occurrence data were collected for the two atlases. We generated all possible combinations of these variables, containing a minimum of three and a maximum of five variables. This resulted in 182 possible combinations. Of these, 10 combinations were discarded as they did not contain both a temperature and precipitation variable. We further discarded 139 combinations after tests for collinearity revealed pairwise correlations between variables of $r > 0.7$[70] (Supplementary Fig. S9). This left 33 potential variable combinations, which were used to build Generalised Additive Models (GAMs) for each of the 378 species. For more details on the fitting of GAMs, see below. We then

ranked the 33 variable combinations using AIC to identify the best performing set of bioclimatic variables for each species (Supplementary Fig. S10). The top set of predictor variables for each species was then used to fit species-specific SDMs, see below. To test how robust our results were to the selection of climatic variables, we also fitted SDMs using the best performing set of climatic variables across all species. To identify this set, we tallied the number of times that each of the 33 variable combinations appeared in the top quartile of candidate sets across all species. This set included mean annual precipitation, seasonality of precipitation, seasonality of temperature, and maximum temperature of the warmest month. This combination was in the highest performing quartile for 85% of the species modelled. We found no substantial differences in the overall results using the different set of climatic predictors (Supplementary Figs. S7 and S8).

To model the relationship between the 1968–1988 bioclimatic variables and the 1985–1988 species' ranges, we used an ensemble SDM modelling framework, combining four widely applied techniques. To provide contrast we used a parametric approach, Generalised Linear Models (GLMs), a semi-parametric approach, GAMs, and two machine-learning approaches, Generalised Boosted Regression Models (GBMs; also referred to as Boosted Regression Trees, BTRs) and Random Forests (RFs). These methods have all been shown to produce models that perform well when used in an ensemble SDM approach[71,72]. Details on the individual modelling approaches and methods to account for spatial autocorrelation (SAC) can be found below. When fitting a model, nine of ten cross-validation sampling blocks were used as the training data set, with model fit assessed using the Area Under the Curve (AUC) of the receiver operating characteristic (ROC) plot[73,74] on the omitted block. As all sampling blocks cover a similar range of bioclimatic data, this method ensures that a similar range of data is used for both testing and training models, whilst also ensuring that the testing data are spatially segregated from the training data (Supplementary Fig. S11). This method has been shown to perform well at a large scale, minimising the influence of SAC whilst allowing models to capture complex spatial processes[71]. By sequentially omitting each of the ten blocks, fitting the model to the remaining nine blocks and testing the performance on the omitted block, ten models were fitted for each of the four modelling techniques. This resulted in 40 models for the breeding ranges for all 378 species. To assess model fit, the median AUC calculated for the omitted blocks was taken across the ten models for each of the four modelling techniques for each of the 378 species.

**Spatial Autocorrelation (SAC).** To account for spatial autocorrelation (SAC) when modelling species' climatic niches, we used a 'blocking' method[71,72], whereby we split the data into ten sampling blocks based on ecoregions ([75], http://www.worldwildlife.org/science/data). SAC occurs when proximate samples show a greater degree of similarity due to distance-related biological processes and spatially structured environmental processes[76]. Failure to account for SAC influences both coefficients and inference in statistical analyses through (1) the violation of the independence assumption and, (2) auto-correlated residuals and hence inflation of type 1 errors[77]. To create the ten sampling blocks, first we classified each non-contiguous area of an ecoregion within the area of study as a separate sampling unit; these sampling units were then grouped into ten blocks so that the mean bioclimate was similar across all blocks, but each block covered the full range of bioclimates within the area of study[72,78] (see Supplementary Fig. S11)

**Generalised Linear Models (GLM).** GLMs[79] were used to fit up to, and including, second order polynomial relationships between the three relevant bioclimatic variables and individual species occurrence. For each species, after omitting one sampling block for model evaluation, nine models (3 bioclimatic variables ^ 2 polynomial degrees = 9 combinations) were fitted to the remaining nine blocks. AUC was then used

to assess the model fit using the excluded block of data. This procedure was repeated excluding each of the ten data sampling blocks sequentially. The combination of polynomial terms for each bioclimatic variable that maximised AUC in each of the ten repeated model fittings was then used to fit a final set of ten models, with each final model fitted to nine blocks of data and evaluated using AUC on the omitted block.

**Generalised Additive Models (GAM).** Relationships between bioclimatic variables and species occurrence were modelled using thin-plate regression splines. Models were fitted to nine blocks of data, after omission of one sampling block for model evaluation using AUC, and the process repeated until each of the ten sampling blocks had been sequentially omitted. These regressions were fitted as a Bernoulli response, using a logit link, and utilised the 'gam' function in the 'mgcv' R package[80].

**Generalised Boosting Methods (GBM).** Generalised boosted models, a machine learning technique, sequentially builds many simple regression trees, which are then combined to optimise predictive performance[81]. This technique requires the user to set three parameters: learning rate (lr; also known as the shrinkage parameter) determines how much each tree contributes to the final model; tree complexity (tc) controls the number of nodes within a tree; and the number of trees (nt) that are to be retained in the final model. We used a cross validation approach to optimise these parameters for each species. Initially, omitting one block at a time, we fitted a model to the remaining nine blocks using an lr of 0.001, an nt of 5000 whilst allowing tc to vary between one and four. The value of tc that returned the minimum summed error across all blocks from a cross-validation approach was used to fit a final set of ten models.

**Random Forests (RF).** Random forests[82,83], are a classification and regression tree (CART) approach, which draws bootstrap samples and a subset of predictors to construct multiple classification trees[84]. This method requires the user to set two parameters; the number of trees (nt) that will constitute the final model and the number of variables randomly sampled as candidates at each split (mtry). We initially set mtry to vary between one and three and then fitted an RF model with 1000 trees to the data after sequentially omitting one block. We assessed the fit of the model on the omitted block using AUC. We then added 500 trees to the model and reassessed AUC. This process was repeated until any improvement in the value of AUC, as a result of the additional trees, was <1%. The values of mtry and nt that maximised mean AUC across the ten blocks of omitted data were used to fit the final ten models.

We applied the 40 SDMs for each species (10 block models x four modelling techniques) to the mean bioclimatic data from the CRU TS 3.25 0.5° dataset[67] for the two time periods (1968–1988 and 1997–2017) which each represented a climatic window in the period preceding and including the two data collection periods. For each species for each period, we took the weighted mean predicted probability of occurrence, weighted by model performance, for each grid cell from the 40 SDMs as a measure of climate suitability. SDMs for all species performed well (mean AUC = 0.94 S.D. ± 0.06, Supplementary Table S1). For each species and for each grid cell we calculated a measure of baseline (referred to in the text as initial) climate suitability by taking the median across all projections from the 40 SDMs for the time period 1968–1988. We also calculated the change in climate suitability for a species in a grid cell by subtracting the median 1968–1988 predicted climate suitability from the median 1997–2017 predicted climate suitability. We take the mean probability of occurrence as, unlike threshold approaches, it does not degrade the information available in our predictions[85]. This climate-only modelling does not produce a realistic projection of change, but it allows us to determine the extent

to which species are being restricted by climate alone. Here we present changes in climate suitability across Europe, alongside plots of changes in ranges between 1985–1988 and 2013–2017, for four species of breeding birds, selected for illustrative purposes (Fig. 3).

To test if predictions of species' responses to environmental change can be improved by accounting for changes in land cover as well as climate, we fitted an additional set of SDMs using both climate and land cover variables. Alongside the climate variables described above, these models included eight aggregated land cover types derived from data obtained from the European Space Agency Climate Change Initiative (ESA CCI, https://www.esa-landcover-cci.org/?q= node/1). Land cover data are described in more detail below. Models were fitted using land cover data from 1992 following the same procedure as above. They were then applied to land cover data for both 1992 and 2015 to predict the changes in species' ranges that would be expected given observed changes in climate and land cover. Although these measures do not fully encompass the period between the two atlases, they are, as far as we are aware, the best available data at this scale.

**Land cover suitability.** To allow us to partition the effect of climate and land cover in our analyses of colonisation and extinction events, we calculated separate measures of land cover suitability and change in land cover suitability in addition to our measures of baseline and change in climate suitability. The land cover data obtained from the ESA CCI, are a global data set available at a resolution of 300 m and consist of 24 annual maps of land cover, comprising 22 land cover classes from 1992–2015. We collated data on each species' primary habitat association, following the Mapping and Assessment of Ecosystems and their Services classification[86]. We then aggregated the 22 land cover types of the spatial land cover data into nine groups that broadly coincide with the classifications used to define species' primary habitat associations[86]. Next, we intersected the land cover types with the same 50 × 50 km grid as the breeding bird atlas data and calculated the area of each of the nine land cover types in each grid cell. For a measure of habitat diversity, we calculated the diversity of land cover types within each grid cell of the aggregated 1992 land cover map using Shannon's diversity index[87]. Using species' primary habitat associations and the aggregated land cover maps, we then calculated the proportion of land cover within each grid cell classified as suitable for a given species in both the 1992 and 2015 maps. We used the amount of suitable land cover for each species in 1992 as a measure of the baseline extent of favourable land cover. As a measure of change in the extent of favourable land cover, we took the difference in the amount of suitable land cover between 1992 and 2015 for each species. To test the robustness of our analysis to the aggregation of land cover types, we repeated our analysis using three different class aggregations. Our results were robust to the class aggregation used (Supplementary Figs. S7, and S8). For the analysis presented here, we use the mean extent of favourable land cover and the change in the extent of favourable land cover within a grid cell from across the three class aggregations.

**Altitudinal range.** The altitudinal range of each grid cell was derived from the ETOPO2 global dataset (available at 1 × 1 km resolution; http://www.ngdc.noaa.gov/mgg/global/etopo2.html).

**Area of protected land.** As a proxy for protection status, we calculated the total area of land within each grid cell classified under the International Union for Conservation of Nature (IUCN) protected area categories I–VI using the World Database on Protected Areas (WDPA: https://www.protectedplanet.net/).

### Evaluating species' range shifts
First, we examined the observed shifts in species' ranges between the two time periods (1985–1988 and 2013–2017). For each species, we calculated the great-circle distance and direction between the centre of gravity (COG) of their observed 1985–1988 range and the COG of their observed 2013–2017 range. For each species, we then calculated the great-circle distance and direction between the COG of the 1970–1990 predicted climate suitability layer and the COG of the 1995–2015 predicted climate suitability layer. This provided a measure of the distance and direction of the observed and predicted shift in species' climatic niches over the 30-year period (Supplementary Figs. S3 and S4). We then compared the observed shifts in species' ranges with the predicted shifts in species' climatic niches using a circular ANOVA[88]. We used a Wilcoxon Signed Rank test to compare the observed and predicted distances of range shifts. We repeated the above analysis to compare observed shifts in species' COG with the predicted shifts in COG from the climate and land cover SDMs. We also assessed the consistency of range shifts between species with different primary habitat associations[86]. We found no consistent direction in the shift of the COG of species ranges between the two periods for any habitat association (Supplementary Fig. S12).

### Assessing predictors of colonisation and extinction events
We used Markov Chain Monte Carlo generalised linear mixed models from the 'MCMCglmm' R package[89] to assess the relationship between the environmental and species' trait covariate sets and the occurrence of colonisation and extinction events. We modelled these two processes separately to enable us to investigate the putative drivers of colonisations and extinctions independent of one another. We defined a colonisation event as a species being recorded as present in a 50 × 50 km grid cell in the 2013–2017 atlas but not in the 1985–1988 atlas. Extinction events occurred where a species was recorded as present in a 50 × 50 km grid cell in the 1985–1988 atlas but not in the 2013–2017 atlas (Supplementary Fig. S5). To control for any variation in sampling effort between the two atlases, we restricted sampling to grid cells where sampling effort has been identified as being more comparable over time[21]. To test how robust our results were to this potential source of bias, we repeated our analysis with grid cells sampled from the full data set. We found no substantial differences in the overall results using the different sampling criteria (Supplementary Figs. S7 and S8). We assumed a binomial distribution, with individual colonisations or extinctions recorded as 'successes' and 'failures' drawn at random depending on the modelled response. For the models of colonisations, we took a random sample of cells from those that a species has never occupied (a *failed* colonisation), equal to the number of colonisation events for that species. As failure to colonise grid cells further away from a species' range was more likely due to dispersal limitations, we also fitted models using a weighted sample of *failures*. The weight was assigned as 1/distance to the nearest continually occupied cell, so that cells closer to a continually occupied cell were given greater weight than those further away. We found no substantial differences between models fitted with weighted or unweighted samples of *failures* (Supplementary Figs. S7 and S8). In the extinction models, we took a random sample of cells from those that a species has continuously occupied, equal to the number of extinction events for that species. The sampling of extinction 'failures' was non-weighted. To account for any potential sampling bias within our results, we repeated this sampling process ten times and repeated the model fitting process described below for all ten absence samples.

MCMCglmm uses a Bayesian approach to fitting generalised linear mixed models. It can account for the non-independence between species that can arise from common ancestry by including a phylogenetic variance–covariance matrix as a random effect. To account for the repeated measurement of species and grid cells, and potential variation in sampling effort and methods between countries, we included these terms as additional random effects. For this, each grid cell was assigned to a country based on which country the grid cell overlapped most with. In addition to the random effects of phylogeny, species, grid cell ID and country, we also included the suite of trait parameters and

environmental covariates detailed above. Finally, to investigate the importance of the spatial structure of populations on species' range dynamics, we also fitted models with and without the distance to the nearest continually occupied grid cell and the distance of a grid cell to the species COG in the first atlas as fixed effects. All continuous predictor variables were standardised using z-transformations. Predictor variables were also checked for collinearity, with no pair of variables having an absolute correlation >0.7. To test for potential interactions between the terms for climate and land cover suitability and change in these variables, we refitted our models including various iterations of these interactions. We found these interaction terms to have a small, non-significant effect (Supplementary Figs. S7 and S8). Models were fitted using non-informative priors with an inverse Wishart distribution ($V = 1$, $v = 0.002$). Model outcomes were insensitive to the specification of the non-informative priors. We ran the model for 220,000 iterations, with a burn-in period of 20,000 and a sampling interval of 200. Approximately 1000 independent samples were generated for each model. We used Gelman–Rubin statistics and diagnostic plots to check for convergence of model chains and the independence of samples.

To account for potential uncertainty in the phylogenetic trees, we randomly selected ten trees from birdtree.org[90] and fitted our models to each of these trees. This resulted in 100 MCMCglmms for each of the colonisation and extinction models (ten phylogenies x ten random absence samples). We then combined the posterior outputs of the resulting models to provide estimates of model coefficients that incorporated uncertainty from both phylogeny and absence sampling. For each model, we calculated the percentage of variance explained by the fixed effects and assessed model performance using marginal $R^2$, following the methods described in Nakagawa & Schielzeth[91]. All analyses were performed in R version 3.6.1[92].

### Reporting summary

Further information on research design is available in the Nature Portfolio Reporting Summary linked to this article.

## Data availability

The data from the first European Breeding Bird Atlas are available from GBIF (https://doi.org/10.15468/adtfvf). The data from the second European Breeding Bird Atlas area available for free from https://ebba2.info/data-request/. Registration will be required for data download. CRU climate data are available from https://crudata.uea.ac.uk/cru/data/hrg/. ESA CCI land cover data are available from https://www.esa-landcover-cci.org/?q=node/1. ETOPO2 altitude data are available from http://www.ngdc.noaa.gov/mgg/global/etopo2.html. WDPA data are available from (https://www.protectedplanet.net/). The species trait data compiled in this study are provided in the Supplementary Information. Source data are provided as a Source Data file. Source data are provided with this paper.

## Code availability

Code to carry out analyses[93] is publicly available on https://github.com/christinehoward399/.

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

## Acknowledgements

This work was supported in large part by funding to S.G.W. from NERC (NE/T001038/1) and from a DU Seedcorn Grant (054_19-20), which supported S.G.W. and C.H. to work on the project. A.L. and E.M. were funded by the Academy of Finland (project 323527, 329251). In addition, the research has been funded through the 2017–2018 Belmont Forum and BiodivERsA joint call for research proposals, under the BiodivScen ERA-Net COFUND programme, and with the funding organisations Academy of Finland (Helsinki: 326338) and the National Science Foundation (CLO, ICER-1927646). A.M.O. was supported by the Spanish Government through the Juan de la Cierva fellowship program—IJCI-2016-30349 and partially supported by the project GREENRISK (PID2020-119933RB-C22). L.B. was partially funded by MCIN/AEI/10.13039/501100011033 through the projects SPEAR (PCI2022-135056-2) and CEX-2018-000828-S "Centro de Excelencia Severo Ochoa". D.Z. acknowledges support from the German Science Foundation (DFG, grant no. ZU 361-1/1). P.M. is funded by the Research project implemented under the National Recovery and Resilience Plan (NRRP), Project title "National Biodiversity Future Center -NBFC". CUP J33C22001190001. We thank Mark Eaton and Ruud Foppen for their contributions to this work through the EBCC Atlas Steering Committee. Finally, we would like to thank the European Bird Census Council (EBCC), its partner organisations and the many thousands of ornithologists who contributed to the data collection.

## Author contributions

C.H., E.M., A.M.O., P.M., L.B., A.L and S.G.W designed the study, performed the analyses, drafted, and contributed to subsequent versions of the manuscript. A.J., and C.M.B., provided additional analytical and design advice. A.A., K.A., Vi.A., Vo.A., D.E.B., H.G.B., T.B., K.A.B., I.J.B., B.B., B.C., T.C., J.C.D.M., V.D.M., N.F., L.F., B.G., C.G., S.H., C.I., M.J., M.K., V.K., P.K., D.K., T.K., D.L., A.L., Q.M., To.M., Ti.M., B.M., K.N., D.N., I.J. Ø., J.-Y.P., C.P., D.P., D.R., S.R., D.Z.R., L.R., Th.S., D.S., P.S., J.S., K.S., S.S., I.S., C.S., E.S., Ti.S., N.T., D.U., C.A.M.v.T., M.V., T.V., A.V., O.V., P.V., T.W., and D.Z., contributed to subsequent versions of the manuscript and/or contributed data.

## Competing interests

The authors declare no competing interests.

## Additional information

Christine Howard [1,65] ✉, Emma-Liina Marjakangas [2,65], Alejandra Morán-Ordóñez [3,4,65], Pietro Milanesi [5,6], Aleksandre Abuladze[7], Karen Aghababyan [8], Vitalie Ajder[9,10], Volen Arkumarev[11], Dawn E. Balmer [12,13], Hans-Günther Bauer[13,14], Colin M. Beale [15,16], Taulant Bino[17], Kerem Ali Boyla[18], Ian J. Burfield[19], Brian Burke[20], Brian Caffrey[20], Tomasz Chodkiewicz [21,22], Juan Carlos Del Moral[23], Vlatka Dumbovic Mazal[24], Néstor Fernández [25,26], Lorenzo Fornasari [27], Bettina Gerlach [28], Carlos Godinho [29], Sergi Herrando [3,13,30], Christina Ieronymidou[31], Alison Johnston [32], Mihailo Jovicevic[33], Mikhail Kalyakin[13,34], Verena Keller [5,13], Peter Knaus [5], Dražen Kotrošan[35], Tatiana Kuzmenko [36], Domingos Leitão[37], Åke Lindström [38], Qenan Maxhuni [39], Tomaž Mihelič[40], Tibor Mikuska[41], Blas Molina[23], Károly Nagy[42], David Noble[12,13], Ingar Jostein Øien [43], Jean-Yves Paquet[44], Clara Pladevall[45], Danae Portolou [46], Dimitrije Radišić [47], Saša Rajkov [48], Draženko Z. Rajković [48], Liutauras Raudonikis[49], Thomas Sattler [5], Darko Saveljić[50], Paul Shimmings[43], Jovica Sjenicic[35,51], Karel Šťastný[52], Stoycho Stoychev[11], Iurii Strus [53], Christoph Sudfeldt[28], Elchin Sultanov[54], Tibor Szép [42,55], Norbert Teufelbauer [56], Danka Uzunova[57], Chris A. M. van Turnhout[58,59], Metodija Velevski [57], Thomas Vikstrøm[60], Alexandre Vintchevski[61], Olga Voltzit [34], Petr Voříšek [13,62], Tomasz Wilk[22], Damaris Zurell [63], Lluís Brotons[3,4,13,64,66], Aleksi Lehikoinen [2,13,66] & Stephen G. Willis [1,66] ✉

[1]Conservation Ecology Group, Department of Biosciences, Durham University, South Road, Durham DH1 3LE, UK. [2]The Helsinki Lab of Ornithology, Finnish Museum of Natural History, University of Helsinki, Helsinki, Finland. [3]Ecological and Forestry Applications Research Centre (CREAF), 08193 Cerdanyola del Vallès, Spain. [4]Forest Science and Tecnology Centre (CTFC), Carretera vella de Sant Llorenç de Morunys km 2, 25280 Sant Llorenç de Morunys, Spain. [5]Swiss Ornithological Institute, Seerose 1, 6204 Sempach, Switzerland. [6]Department of Biological, Geological and Environmental Sciences (BiGeA), University of Bologna, Via F. Selmi 3, 40126 Bologna, Italy. [7]Institute of Zoology, Ilia State University, Kakutsa Cholokashvili Ave 3 / 5, Tbilisi 0162, Georgia. [8]BirdLinks Armenia (former TSE—Towards Sustainable Ecosystems) NGO, 87b Dimitrov, apt 14, Yerevan, Armenia. [9]Society for Birds and Nature Protection, Leova, Republic of Moldova. [10]Moldova State University, A.Mateevici str. 60, Chişinău, Republic of Moldova. [11]Bulgarian Society for the Protection of Birds/ BirdLife Bulgaria, Sofia 1111, Yavorov complex, bl. 71, en. 1, ap. 1, Sofia, Bulgaria. [12]British Trust for Ornithology, The Nunnery, Thetford, Norfolk IP24 2PU, UK. [13]Atlas Steering Committee, European Bird Census Council, Na Bělidle 34, CZ-150 00 Prague 5, Czech Republic. [14]Max-Planck Institute of Animal Behaviour, Am Obstberg 1, 78315 Radolfzell, Germany. [15]York Environmental Sustainability Institute, University of York, York YO10 5NG, UK. [16]Department of Biology, University of York, YO10 5DD York, UK. [17]Albanian Ornithological Society, Rr. "Vaso Pasha", Nd. 4, Apt. 3, 1004 Tirana, Albania. [18]WWF Turkey, Büyük Postane Caddesi No: 19 Kat: 5, 34420Bahçekapı-Fatih, İstanbul, Turkey. [19]BirdLife International, David Attenborough Building, Pembroke Street, Cambridge CB2 3QZ, UK. [20]BirdWatch Ireland, Unit 20, Block D, Bullford Business Campus, Kilcoole, Greystones, County Wicklow, Ireland. [21]Museum and Institute of Zoology, Polish Academy of Sciences, Wilcza 64, 00-679 Warszawa, Poland. [22]Polish Society for the Protection of Birds, Odrowąża 24, 05-270 Marki, Poland. [23]Sociedad Española de Ornitología (SEO/BirdLife), Melquiades Biencinto, 34, 28053 Madrid, Spain. [24]Institute for Environment and Nature, Ministry of Economy and Sustainable Development, Radnicka cesta 80, 10 000 Zagreb, Croatia. [25]German Centre for Integrative Biodiversity Research (iDiv) Halle-Jena-Leipzig, Leipzig, Germany. [26]Inst. of Biology, Martin Luther Univ. Halle-Wittenberg, Halle, Germany. [27]Associazione FaunaViva, Via Fumagalli 6, 20143 Milano, Italy. [28]DDA—Federation of German Avifaunists, An den Speichern 2, D-48157 Münster, Germany. [29]MED—Mediterranean Institute for Agriculture, Environment and Development; LabOr—Laboratório de Ornitologia Universidade de Évora Pólo da Mitra, Apartado 94, 7002-774 Évora, Portugal. [30]Catalan Ornithological Institute, Natural History Museum of Barcelona, Plaça Leonardo da Vinci 4–5, 08019 Barcelona, Spain. [31]BirdLife Cyprus, P.O. Box 12026 Nicosia 2340, Cyprus. [32]Centre for Research into Ecological and Environmental Modelling, University of St Andrews, St Andrews, UK. [33]Pro Natura, Donji Crnci bb, 81412 Spuž, Montenegro. [34]Zoological Museum of Lomonosov Moscow State University, Bolshaya Nikitskaya Str., 2, Moscow 125009, Russia. [35]Ornithological society "Naše ptice", Semira Frašte 6, 71 000, Sarajevo, Bosnia and Herzegovina. [36]Ukrainian Society for the Protection of Birds, P.O. Box 33, Kyiv 01103, Ukraine. [37]Sociedade Portuguesa para o Estudo das Aves, Av. Almirante Gago Coutinho, 46A, 1700-031 Lisboa, Portugal. [38]Department of Biology, Lund University, Lund, Sweden. [39]Kosovo Ornithological Society, Str. Hysni Gashi no. 28, Kalabri, 10 000 Prishtinë, Republic of Kosovo. [40]DOPPS—BirdLife Slovenia, Tržaška c. 2, SI, 1000 Ljubljana, Slovenia. [41]Croatian Society for Birds and Nature Protection, Gundulićeva 19a, HR-31000 Osijek, Croatia. [42]MME BirdLife Hungary, 1121 Költő u. 21, Budapest, Hungary. [43]BirdLife Norway. Sandgata 30b, NO-7012 Trondheim, Norway. [44]Natagora, Traverse des muses 1, 5000 Namur, Belgium. [45]Andorra Research + Innovation, Av. Rocafort 21-23, AD600 Sant Julià de Lòria, Andorra. [46]Hellenic Ornithological Society / BirdLife Greece, Agiou Konstantinou 52, Athens 10437, Greece. [47]University of Novi Sad, Faculty of Sciences, Department of Biology and Ecology, Trg Dositeja Obradovića 3, Novi Sad 21000, Serbia. [48]Center for Biodiversity Research, Maksima Gorkog 40/3, 21000 Novi Sad, Serbia. [49]Lithuanian Ornithological Society, Naugarduko st. 47-3, LT-03208 Vilnius, Lithuania. [50]Environmental Protection Agency of Montenegro, IV proleterske 19, 81000 Podgorica, Montenegro. [51]Society for Research and Protection of Biodiversity, Mladena Stojanovica 2, 78 000 Banja Luka, Bosnia and Herzegovina. [52]Czech University of Life Sciences, Faculty of Environmental Sciences, Dept. of Ecology, Kamýcká 129, 165 21 Prague 6—Suchdol, Prague, Czech Republic. [53]Nature reserve "Roztochya", Sichovyh Striltsiv 7, 81070 Ivano-Frankove, Ukraine. [54]Azerbaijan Ornithological Society, M. Mushfiq 4B, ap.60, Baku AZ1004, Azerbaijan Republic. [55]University of Nyíregyháza, 4400 Sóstói út 31/b, Nyíregyháza, Hungary. [56]BirdLife Österreich, Museumsplatz 1/10/8, A-1070 Wien, Austria. [57]Macedonian Ecological Society, Blvd. Boris Trajkovski Str. 7, 9a, Skopje, N, Macedonia. [58]Sovon—Dutch Centre for Field Ornithology, Nijmegen, The Netherlands. [59]Radboud Institute for Biological and Environmental Sciences, Radboud University, Nijmegen, The Netherlands. [60]Dansk Ornitologisk Forening (DOF—BirdLife DK), Copenhagen, Denmark. [61]TAA "Dzikaja pryroda", Parnikovaya Street 11, office 4, Minsk 220114, Belarus. [62]Czech Society for Ornithology, Na Bělidle 34, 15000 Prague 5, Czechia. [63]Institute for Biochemistry and Biology, University of Potsdam, Potsdam, Germany. [64]CSIC, Cerdanyola del Vallès 08193, Spain. [65]These authors contributed equally: Christine Howard, Emma-Liina Marjakangas, Alejandra Morán-Ordóñez. [66]These authors jointly supervised this work: Lluís Brotons, Aleksi Lehikoinen, Stephen G. Willis. ✉e-mail: christine.howard@durham.ac.uk; s.g.willis@durham.ac.uk

