## [Peer Review File · Nature Communications]

Reviewer comments, first round review

Reviewer #3 (Remarks to the Author):

This manuscript highlights the disconnect between modeled and observed range changes in birds. The authors use two approaches: first, they show misalignment in change in geographic range centroids, and second, they show that climate change, landcover change, and species traits do not predict colonization and extinction events. The manuscript is well-written and the models are appropriately applied and interpreted. The major findings of the paper have been shown before, both in terms of climate only SDMs not aligning with observations (ref below) and SDMs not capturing colonization and extinction (refs cited). However, this paper does add some novel aspects, and does so rigorously and at a continental scale.

Overall, the methodological approaches are sound. Models are based on presences and reasonably-inferred absences, and incorporated spatial blocking. Robust sensitivity analyses are shown in S6 and S7. The authors responded thoroughly to comments from the previous three reviewers.

I have a few suggestions for the authors to consider, which could contribute to the novelty of the work. The SDMs are climate-only models, and the results from this study align closely with a new paper demonstrating a lack of congruence between observed and predicted change in range centroids in North American birds (Huang et al. 2023 *Science of the Total Env.* <http://dx.doi.org/10.1016/j.scitotenv.2022.159603>). The authors calculated land cover suitability and change in land cover suitability in a guild-specific manner that strikes me as just about the best attempt one could make at a continental scale. These covariates were used along with climate suitability derived from the climate-only SDM in the modeling of colonization and extinction, but I wonder if it's worth refitting SDMs with both climate and landcover variables (I would leave the col/ext analysis to the climate-only SDMs, since it's valuable there to partition effects of climate and landcover). Right now the take-away of the manuscript is that climate only SDMs are likely unreliable, and it would be interesting to test if incorporating land cover change improves predicted range changes (and would substantively extend this work beyond the Huang et al. paper). If so, it would highlight the importance of appropriately quantifying and including land cover, and if not, it would provide a stark showing that even climate+landcover SDMs are worrisome in their reliability.

Similarly, for the models of colonization and extinction, the most important variable by far is the distance from the nearest continuously occupied cell. Unlike all the other variables, this is something that one doesn't know when doing forward-looking projections. I think it would be useful to compare predictive ability with and without this variable. If climate change, landcover change, and species traits collectively are only weakly related to range change, what do the authors have in mind when they refer to 'integrative, multi-driver approaches for more robust forecasting' (last line of abstract)?

Line comments:

109: untested is a little strong given the multiple previous studies

458: not clear from this section if this is a static variable or if you evaluated how change in land cover suitability affected colonization and extinction. Small edits here could make it clear you did capture change.

Reviewer #4 (Remarks to the Author):

It seems like a fundamental assumption of this approach is that the lack of a relationship between changing climate suitability and observed range shifts depends on the ability of the modeling framework and chosen climate variables to accurately capture the climatic niche for individual species. If not, it is possible that the absence of an association between a shift in climate

suitability and observed range change is less about the lack of an important climate relationship, but that the climate variables being used do not capture the climatic niche of the species. This is an inherent problem when dealing with hundreds of species and was clearly documented in two opposing studies regarding the importance of choosing relevant climate variables for understanding the role of climate in constraining species distributions; the original paper by Beale et al. (2008) and the follow up paper by Araujo et al. 2009 that came to contradictory conclusions. I believe the same issue applies here with the use of three bioclimatic variables to describe climate niches for hundreds of bird species (despite reporting AUC values > 0.8 for many species; which is not surprising at this scale and given certain problems associated with relying on AUC). I also felt like the justification for the three climate variables [L393-395] was lacking and didn't not adequately cover issues of multicollinearity of the ecological justification necessary for all these species. For example, the GDD cutoff of 5 degrees was not explained. The studies cited as further justification for these variables for European bird abundance were addressing different questions, species, and temporal extents.

CM Beale, JJ Lennon, A Gimona, Opening the climate envelope reveals no macroscale associations with climate in European birds. *Proc Natl Acad Sci USA* 105, 14908–14912 (2008).

MB Araujo, W. Thuiller, and N.G. Toccoz. Reopening the climate envelope reveals macroscale associations with climate in European birds. *Proc Natl Acad Sci* 106(16) E45-E46 (2009)

Although I applaud the effort and extent of this work, I am struggle with whether the broader conclusions - that changes in climate suitability is a poor predictor of observed range changes - is sufficiently different from many previous studies on the subject performed at continental scales. I see that this issue was raised by previous reviewers who list many of these studies. I believe most in the field are well aware that climate-only models are not good predictors of future range shifts, and as such, many studies are very careful to lay out the assumptions, present caveats, carefully choose ecologically-relevant climate variables, and incorporate information on land cover or land use change.

Additional comments:

L159: I believe there are many studies (both for birds and other taxa) that explore species-level analyses of expected climate-driven range change (e.g., most of Josh Lawler's work and many others)

L180: I would include phenotypic plasticity in addition to variation

L386-388: Could there be issues of fusing polygon range data and atlas data? This seems to pose many problems with scale or resolution and incorporating the environmental data.

L398-400: I was confused by the range of dates on the climate data. Why 1970-1990 for an atlas period that covered 1985-1988? Similarly, why 1995-2015 for an atlas period of 2013 to 2017? I know there is a decision to try and approximate the broader climate conditions but 1) the first climate period includes two years of climate that the species didn't even experience and the second period did not include two years of atlasing that the species were exposed to, and 2) this ignores any regional climate conditions (e.g., droughts, heat waves) that were being experienced during the atlas periods.

Response to reviewers

Reviewer Comments	Our response
Reviewer #3:	
This manuscript highlights the disconnect between modeled and observed range changes in birds. The authors use two approaches: first, they show misalignment in change in geographic range centroids, and second, they show that climate change, landcover change, and species traits do not predict colonization and extinction events. The manuscript is well-written and the models are appropriately applied and interpreted. The major findings of the paper have been shown before, both in terms of climate only SDMs not aligning with observations (ref below) and SDMs not capturing colonization and extinction (refs cited). However, this paper does add some novel aspects, and does so rigorously and at a continental scale. Overall, the methodological approaches are sound. Models are based on presences and reasonably-inferred absences, and incorporated spatial blocking. Robust sensitivity analyses are shown in S6 and S7. The authors responded thoroughly to comments from the previous three reviewers.	We thank the reviewer for the positive comments.
I have a few suggestions for the authors to consider, which could contribute to the novelty of the work. The SDMs are climate-only models, and the results from this study align closely with a new paper demonstrating a lack of congruence between observed and predicted change in range centroids in North American birds (Huang et al. 2023 Science of the Total Env. http://dx.doi.org/10.1016/j.scitotenv.2022.159603). The authors calculated land cover suitability and change in land cover suitability in a guild-specific manner that strikes me as just about the best attempt one could make at a continental scale. These covariates were used along with climate suitability derived from the climate-only SDM in the modeling of colonization and extinction, but I wonder if it's worth refitting SDMs with both climate and landcover variables (I would leave the col/ext analysis to the climate-only SDMs, since it's valuable there to partition effects of climate and landcover). Right now the take-away of the manuscript is that climate only SDMs are likely unreliable, and it would be interesting to test if incorporating land cover	Thank you for pointing out this new paper, which we have cited in our introduction (Line 163). We also thank the reviewer for suggesting that we refit our SDMs including land cover. We have now refitted out SDMs including the eight aggregated ESA land cover types, described in our methods. We found that the inclusion of land cover variables did not significantly improve our ability to predict observed changes in species' ranges (Figure 1c.). We have also updated our methods (Lines 479-488), results (Lines 213 -220) and discussion (Lines 337-339) accordingly. We believe that this further analysis provides the additional novel aspect requested by the reviewers, differentiating our manuscript from those papers highlighted by the reviewers that have focussed solely on the effects of climate.

change improves predicted range changes (and would substantively extend this work beyond the Huang et al. paper). If so, it would highlight the importance of appropriately quantifying and including land cover, and if not, it would provide a stark showing that even climate+landcover SDMs are worrisome in their reliability.	
Similarly, for the models of colonization and extinction, the most important variable by far is the distance from the nearest continuously occupied cell. Unlike all the other variables, this is something that one doesn't know when doing forward-looking projections. I think it would be useful to compare predictive ability with and without this variable. If climate change, landcover change, and species traits collectively are only weakly related to range change, what do the authors have in mind when they refer to 'integrative, multi-driver approaches for more robust forecasting' (last line of abstract)?	Thank you for this suggestion, which we have now implemented. We found that models of colonisation and extinction events fitted without this variable performed substantially worse than models fitted with it (Lines 256 -263). This result further highlights the role of interactions between spatially structured populations in moderating species' range shifts. We believe that accounting for these metapopulation processes into analyses of potential range shifts, would allow for more robust forecasting. We have clarified this in lines 341-344.
109: untested is a little strong given the multiple previous studies	Apologies, we have changed this to 'uncertain' (Line 112)
458: not clear from this section if this is a static variable or if you evaluated how change in land cover suitability affected colonization and extinction. Small edits here could make it clear you did capture change.	Our apologies. We have now edited this section to make it clearer that we were also measuring change in land cover suitability (Lines 505 - 511).
Reviewer #4:	
It seems like a fundamental assumption of this approach is that the lack of a relationship between changing climate suitability and observed range shifts depends on the ability of the modeling framework and chosen climate variables to accurately capture the climatic niche for individual species. If not, it is possible that the absence of an association between a shift in climate suitability and observed range change is less about the lack of an important climate relationship, but that the climate variables being used do not capture the climatic niche of the species. This is an inherent problem when dealing with hundreds of species and was clearly documented in two opposing studies regarding the importance of choosing relevant climate variables for understanding the role of climate in constraining species distributions; the original paper by Beale et al. (2008) and the follow up paper by Araujo et al. 2009 that came to	Thank you for raising this point. Ideally, to address this concern we would use climate variables selected by experts with species-specific knowledge to ensure that the most relevant variables were used for fitting SDMs¹. Unfortunately, this information is not available for all 378 species modelled here. We have therefore implemented a variable selection process, following⁵, that we hope addresses the reviewer's concern. We have provided specific details in our methods (Lines 409-441), but briefly summarise our approach here. We preselected eight bioclimatic variables from WorldClim that have been used widely in climatic niche modelling. We then generated 33 combinations of these climate variables with between 3 and 5 variables, where both temperature and precipitation were represented, and pairwise correlations between variables did not exceed 0.7. For each species,

contradictory conclusions. I believe the same issue applies here with the use of three bioclimatic variables to describe climate niches for hundreds of bird species (despite reporting AUC values > 0.8 for many species; which is not surprising at this scale and given certain problems associated with relying on AUC). I also felt like the justification for the three climate variables [L393-395] was lacking and didn't not adequately cover issues of multicollinearity of the ecological justification necessary for all these species. For example, the GDD cutoff of 5 degrees was not explained. The studies cited as further justification for these variables for European bird abundance were addressing different questions, species, and temporal extents. CM Beale, JJ Lennon, A Gimona, Opening the climate envelope reveals no macroscale associations with climate in European birds. Proc Natl Acad Sci USA 105, 14908–14912 (2008). MB Araujo, W. Thuiller, and N.G. Toccoz. Reopening the climate envelope reveals macroscale associations with climate in European birds. Proc Natl Acad Sci 106(16) E45-E46 (2009)	we then used GAMs to establish which combinations of variables produced the best performing model, measured using AIC. The best performing set of climate variables for each species was then used to fit the full set of 40 SDMS (4 modelling methods x 10 sampling blocks). To test how sensitive our models were to the choice of climate variables we also fitted the full SDMs using the single set of climate variables that were the best performing across all 378 species. We found that neither approach, i.e. the individual climate variable set for each species or the overall best performing climate variable set, produced substantially better predictions of observed range shifts. We used the measures of climate suitability derived from the SDMs using the species-specific climate variables to refit our models of colonisation and local extinctions. This has had no substantial impact on our results (Figure 3). We have updated our methods (lines 409 -441) and results (lines 208 - 213) accordingly and have included a figure comparing predicted range shifts from the two approaches in our supplementary materials (Supplementary Figure 3).
Although I applaud the effort and extent of this work, I am struggle with whether the broader conclusions - that changes in climate suitability is a poor predictor of observed range changes - is sufficiently different from many previous studies on the subject performed at continental scales. I see that this issue was raised by previous reviewers who list many of these studies. I believe most in the field are well aware that climate-only models are not good predictors of future range shifts, and as such, many studies are very careful to lay out the assumptions, present caveats, carefully choose ecologically-relevant climate variables, and incorporate information on land cover or land use change.	We have now, in response to a comment by the other reviewer, refitted our SDMs with land cover variables as well as climate variables. This additional analysis has revealed that the inclusion of land cover variables does not substantially improve our ability to predict observed range shifts (Lines 213-220, Figure 1c., and Supplementary Figure S3). As far as we are aware, this is the first time that this has been shown and has substantial implications for species distribution modelling. Our analysis also goes beyond previous analyses of the efficacy of climate only modelling, to show that species' observed range shifts are strongly driven by processes such as proximity to source populations (Figure 2). This result also has important implications for efforts to forecast species responses to environmental change, highlighting the importance of considering metapopulation processes for robust forecasting.
L159: I believe there are many studies (both for birds and other taxa) that explore species-level analyses of expected climate-driven range	The reviewer is correct in that there are many studies exploring species-levels predictions of the expected impacts of climate change. The

change (e.g., most of Josh Lawler's work and many others)	point, however, that we were making on L159, was that there are very few studies that have compared these expected range shifts with observed range shifts. We have reworded this text to ensure that this point is clearer and included additional references to the literature (Lines 161 - 166).
L180: I would include phenotypic plasticity in addition to variation	We have now included this (Line 296).
L386-388: Could there be issues of fusing polygon range data and atlas data? This seems to pose many problems with scale or resolution and incorporating the environmental data.	By including Turkey and North Africa in our SDMS, we can explicitly model the relationship between species occurrence and areas of climate space that may be relevant to range changes in southern and, to some extent, central Europe. Without this approach, we would have to clamp our predictions of species distributions, to the areas of climate space modelled in the first atlas to avoid extrapolation. This would in turn limit our ability to fully explore range change⁴. We agree that range polygon data may be more simplistic than atlas data in some regions of the world. However, a national breeding atlas has been produced for Turkey (published 1995, with underlying data period relevant to EBBA1 monitoring) and the polygons there are based on these point data and hence are more precise than is the case in less well-monitored areas. For North Africa, it is the case for most European breeding species that their ranges occur principally along the coastal fringe and in the limited areas of temperate habitat, such as the Atlas Mountains and the temperate forest of e.g Tunisia. As such, we consider the polygon data for breeding records of 'European' species in these areas is also robust. That is certainly our experience from visiting such areas. With regards to the environmental data, these all derive from global datasets and should in theory be of the same quality in both regions.
L398-400: I was confused by the range of dates on the climate data. Why 1970-1990 for an atlas period that covered 1985-1988? Similarly, why 1995-2015 for an atlas period of 2013 to 2017? I know there is a decision to try and approximate the broader climate conditions but 1) the first climate period includes two years of climate that the species didn't even experience and the second period did not include two years of atlasing that the species were exposed to, and 2)	Our apologies. We have now refitted all SDMs with climate data pertaining to 1968-1988 and made predictions to climate data from 1968-1988 and 1997-2017, which better aligns with the two atlas periods (1985-1988 and 2013-2017). The use of a 20 – 30-year climate normal is the standard practice in this field. However, we do recognise that variables representing typical

this ignores any regional climate conditions (e.g., droughts, heat waves) that were being experienced during the atlas periods.

climatic conditions may not adequately reflect any extremes in regional weather. We have therefore also included climate variables that describe the extremes in temperature and precipitation (maximum temperature of the warmest month, minimum temperature of the coldest month, precipitation of the wettest month, and precipitation of the driest month) in our variable selection process (Line 409 – 441). We hope this addresses the reviewer’s concern.

References

1. Barbet-Massin, M., Thuiller, W. & Jiguet, F. How much do we overestimate future local extinction rates when restricting the range of occurrence data in climate suitability models? *Ecography (Cop.)*. **33**, 878–886 (2010).

Reviewer comments, second round review

Reviewer #3 (Remarks to the Author):

The authors implemented the major suggested changes, conducting new analyses that included land cover change in SDMs and that attempted to predict colonization and extinctions without knowledge of which grid cells remained occupied. Overall, their study provides another rigorous demonstration of the inability of SDMs to predict colonization and extinction across space, and their inclusion of land cover and explicit modeling of colonization and extinction yields a comprehensive set of analyses that extends previous work.

My one comment about the added set of colonization and extinction models is that they are framed in terms of understanding the effects of metapopulation dynamics / spatial structure, and it isn't emphasized that modeling without knowing which cells are continuously occupied would be the norm for any forward-looking study, and therefore predictive ability will be quite low.

Line comments:

116: 'different from expectations'

133: 'fecundity' instead of 'productivity'? Seems like more typical language for birds

162-165: Several of these studies encompassed tens to hundreds of species so it is not correct to state that this has been done for 'a few species'. Of these studies, Rapacciuolo et al. included (static) topography and geology; Briscoe et al. included static land cover and changing NDVI; Sofaer et al. included land use change; Venne and Currie included neighborhood occupancy. I think you need to restate how your work confirms/extends previous findings and refine your statement of what is novel.

220: It doesn't feel like this line about predicting the median shifts well aligns with the values given above, nor with the text above about the significant differences.

351: not sure this last sentence is supported

386: Shannon's diversity?

Reviewer #4 (Remarks to the Author):

I have reviewed your responses to my original concerns (and the concerns posed by the other reviewers). I believe you have addressed my concerns by better matching the climate and atlas data and incorporating land use data. I still have concerns about the broader conclusions regarding the modeling of climate suitability as opposed to future species occurrences, but I believe these kind of analyses are important for testing the underlying assumptions of ecological niche models.

Response to reviewer's comments

Reviewer Comment	Response
Reviewer 3	
The authors implemented the major suggested changes, conducting new analyses that included land cover change in SDMs and that attempted to predict colonization and extinctions without knowledge of which grid cells remained occupied. Overall, their study provides another rigorous demonstration of the inability of SDMs to predict colonization and extinction across space, and their inclusion of land cover and explicit modeling of colonization and extinction yields a comprehensive set of analyses that extends previous work.	We thank the reviewer for their comments and helpful feedback.
My one comment about the added set of colonization and extinction models is that they are framed in terms of understanding the effects of metapopulation dynamics / spatial structure, and it isn't emphasized that modeling without knowing which cells are continuously occupied would be the norm for any forward-looking study, and therefore predictive ability will be quite low.	Thank you for highlighting this point. We have now added a sentence to emphasise that without knowing which areas will be continuously occupied, the norm for any forward-looking study, our ability to predict species' range change will likely be poor (Lines 262 - 264).
116: 'different from expectations'	Now changed as suggested (Line 116)
133: 'fecundity' instead of 'productivity'? Seems like more typical language for birds	Now changed as suggested (Line 133)
162-165: Several of these studies encompassed tens to hundreds of species so it is not correct to state that this has been done for 'a few species'. Of these studies, Rapacciuolo et al. included (static) topography and geology; Briscoe et al. included static land cover and changing NDVI; Sofaer et al. included land use change; Venne and Currie included neighborhood occupancy. I think you need to restate how your work confirms/extends previous findings and refine your statement of what is novel.	We have now reworked these sentences to emphasise that our work goes beyond the previous studies that have demonstrated that SDMs often fail to predict species' range shifts, to assess the extent to which climate change has driven these range shifts in relation to other putative drivers (Lines 161 - 165).
220: It doesn't feel like this line about predicting the median shifts well aligns with the values given above, nor with the text above about the significant differences.	We have now reworded this sentence to better align with the preceding text (Lines 218 - 220).
351: not sure this last sentence is supported	We have deleted this sentence (Lines 352- 353).
386: Shannon's diversity?	For this analysis we used Shannon's richness index, instead of Shannon's diversity index (Line 387).
Reviewer 4	

I have reviewed your responses to my original concerns (and the concerns posed by the other reviewers). I believe you have addressed my concerns by better matching the climate and atlas data and incorporating land use data. I still have concerns about the broader conclusions regarding the modeling of climate suitability as opposed to future species occurrences, but I believe these kind of analyses are important for testing the underlying assumptions of ecological niche models.	We thank the reviewer for their positive comments and for seeing the merit in our work. We understand the concerns that the reviewer has regarding the modelling of climate suitability and that the absence of a relationship may be a consequence of the modelling procedure. We have detailed these methodological limitations in our discussion (Lines 237 - 244).
--	---